# Perennial Baki™ Bean Safety for Human Consumption: Evidence from an Analysis of Heavy Metals, Folate, Canavanine, Mycotoxins, Microorganisms and Pesticides

**DOI:** 10.3390/molecules29081777

**Published:** 2024-04-13

**Authors:** Evan B. Craine, Muhammet Şakiroğlu, Spencer Barriball, Tessa E. Peters, Brandon Schlautman

**Affiliations:** 1The Land Institute, Salina, KS 67401, USA; barriball@landinstitute.org (S.B.); peters@landinstitute.org (T.E.P.); 2Bioengineering Department, Adana Alparslan Türkeş Science and Technology University, Adana 01250, Turkey; msakiroglu@gmail.com

**Keywords:** perennial grain crop, safety, sainfoin, heavy metals, mycotoxins, canavanine

## Abstract

Global food production relies on annual grain crops. The reliability and productivity of these crops are threatened by adaptations to climate change and unsustainable rates of soil loss associated with their cultivation. Perennial grain crops, which do not require planting every year, have been proposed as a transformative solution to these challenges. Perennial grain crops typically rely on wild species as direct domesticates or as sources of perenniality in hybridization with annual grains. *Onobrychis* spp. (sainfoins) are a genus of perennial legumes domesticated as ancient forages. Baki™ bean is the tradename for pulses derived from sainfoins, with ongoing domestication underway to extend demonstrated benefits to sustainable agriculture. This study contributes to a growing body of evidence characterizing the nutritional quality of Baki™ bean. Through two studies, we investigated the safety of Baki™ bean for human consumption. We quantified heavy metals, folate, and canavanine for samples from commercial seed producers, and we quantified levels of mycotoxins, microorganisms, and pesticides in samples from a single year and seed producer, representing different varieties and production locations. The investigated analytes were not detectable or occurred at levels that do not pose a significant safety risk. Overall, this study supports the safety of Baki™ bean for human consumption as a novel pulse crop.

## 1. Introduction

The crops that have sustained humans since the advent of agriculture are facing unprecedented challenges. Volatile and extreme weather events and unsustainable soil loss associated with the unintended consequences of annual establishment and production threaten annual grain cropping systems [1]. Impacts on food and nutritional security are already being seen. Among the top three annual cereal grains, yields have decreased for rice and wheat, and have negligibly increased for corn, resulting in an estimated 0.4%, 0.5%, and 0.7% decrease in consumable food calories from rice, wheat, and maize, respectively [2]. Diverse, perennial grain crop systems have been envisioned as a tractable solution to overcome these challenges [3,4,5,6]. By regrowing year after year, perennial grain crops do not require the removal of vegetation and soil disturbances each year as annual crops do and therefore require fewer inputs and labor than annual crops [7,8,9,10,11]. Recent advances in perennial rice and Kernza^®^ perennial grain, the grain produced from improved intermediate wheatgrass (*Thinopyrum intermedium*), demonstrate the feasibility of perennial grain crops and the potential benefits to environmental, economic, and social sustainability [8,10,12,13,14]. For example, perennial rice saved growers 58.1% of labor and 49.2% of input costs in each regrowth cycle. Over eight harvests during four consecutive regrowth cycles, soil carbon and nitrogen content increased, and perennial rice produced slightly more grain than annual rice during each harvest [14].

Two parallel strategies have been utilized for perennial grain development. Both de novo domestication and wide hybridization rely on wild species for perenniality and other favorable traits. Perennial rice is the hybrid of annual Asian rice (*Oryza sativa*) and its wild African relative (*Oryza longistaminata*) [14]. Kernza is produced from intermediate wheatgrass domesticated for grain production [12,15]. Similar to intermediate wheatgrass, which had been successfully domesticated as a forage crop, sainfoins (*Onobrychis* spp.) are ancient perennial legumes and forage crops being domesticated as a grain legume or pulse crop [16]. Both domesticated and wild *Onobrychis* spp. have advantageous agronomic and ecological characteristics that support diversified uses and adaptations to climate change [17]. Sainfoins are deep-rooted, nitrogen-fixing, drought-tolerant perennials, and they are an excellent forage for ruminants and bees [18,19,20,21]. Therefore, they provided numerous ecosystems services compared to annual legumes.

Domestication as a pulse crop will further contribute to renewed interest in sainfoins with great potential for sustainable agriculture. Baki™ bean is the tradename for pulses derived from sainfoins, an effort led by The Land Institute (Salina, KS, USA) with global partners. The development of any novel crop for human consumption requires a thorough and comprehensive evaluation to demonstrate safety and quality. This is the goal of ongoing Baki™ bean research and development efforts. We previously reported the unique and distinctive features of Baki™ bean compared to other pulse crops [22]. Furthermore, we reported favorable amino acid and fatty acid profiles for Baki™ bean [23]. Building on these insights into Baki™ bean nutritional quality, we present two separate and complementary studies investigating the safety of Baki™ bean for human consumption. The first study investigated representative lots of seed from commercial producers and determined the content of folate, specific elements (e.g., heavy metals), and L-canavanine, a non-proteinogenic amino acid found in certain leguminous plants. The second study investigated seed produced from a single producer in a single year, representing different varieties and production locations, and determined the content of mycotoxins, microorganisms, and pesticides, in addition to macronutrients. These two studies were designed to investigate whether these compounds are present in Baki™ bean and at what levels. Together, they serve as a preliminary study, for which there are not any in the literature, that will support more in-depth analysis in the future.

## 2. Results and Discussion

### 2.1. Study I

Results are presented in Table 1.

#### 2.1.1. Elemental Analysis

Monitoring of specific elements, such as the heavy metals analyzed in this study, is important to ensure that the levels observed do not pose a significant risk to plants or people when consumed. While not strictly defined, heavy metals include elements with specific weights greater than 5 g cm^−3^ [24]. Of the approximately 40 elements in this group, lead, cadmium, arsenic, and mercury are the elements of primary concern due to the risk of toxicity [25]. 

Baki™ bean cadmium content ranged from 0.03 to 0.15 ppm (Table 1). Three samples had values greater than 0.10 ppm. The sample distribution mean (0.08 ppm) was not significantly lower than the maximum level for pulses of 0.1 ppm (t(8) = −1.11, *p* = 0.15). Edirisinghe & Jinadasa (2019) provide an analysis of cadmium content in legumes and cereals in Sri Lanka, reporting a range of mean values from 0.01 ppm for cowpea to 0.04 ppm for sesame [26]. Cadmium accumulation in staple foods can present a considerable risk to public health. For example, cadmium accumulation in rice is considered to be a widespread problem in populations that rely on rice as a primary component of the diet [27]. Therefore, the advised limit for cadmium is 0.2 ppm [28]. Rice cultivation practices, such as irrigation and lime applications, can impact cadmium content in rice [29]. Furthermore, when used, commercial phosphate fertilizers can be a source of cadmium in rice [30] and other agricultural production systems [27]. This is important to consider when developing best practices for Baki™ bean production. 

Barium content had a wide range in Baki™ bean (1.13–9.6 ppm) (Table 1). Baki™ bean had a lower mean value (3.49 ppm) than the value reported by Pearson & Ashmore (2020) for bran cereal flakes (8.9 ppm) [31]. Compared to various categories of foodstuffs consumed by a Spanish population, the observed barium content of Baki™ bean is lower than nuts (5.210 ppm), and higher than fats and oil (2.523 ppm), cereals (0.679 to 1.591 ppm), and pulses (0.462 to 0.693 ppm) [32]. Baki™ bean samples contained arsenic, beryllium, chromium, lead, mercury, and silver below the detection limit; these heavy metals were practically absent (Table 1). For arsenic, the sample distribution mean (0.05 ppm) was significantly lower than the maximum level listed for husked rice (0.35 ppm) (t(8) = −620, *p* < 0.001). For mercury, the sample distribution mean (0.005 ppm) was significantly lower than the lowest maximum level listed for seafood (0.8 ppm) (t(8) = −4731, *p* < 0.001). Finally, the sample distribution mean for lead (0.05 ppm) was significantly lower than the lowest maximum level listed for pulses (0.1 ppm) (t(8) = −502, *p* < 0.001).

Baki™ bean is not yet commercially available and is likely to have low prevalence in diets. Therefore, a low intake level is expected when initially commercialized. Even if consumed at levels comparable to staple crops, Baki™ bean likely poses minimal concern to human health due to levels of cadmium and barium comparable to other foods. However, further studies should monitor cadmium levels, as certain samples did have cadmium content above the maximum level for pulses, and identify contributing factors where possible.

Certain elements, such as zinc and iron, are also classified as heavy metals. They are also known to be essential nutrients required by plants and people. Craine et al. (2023) found the content of iron and zinc in Baki™ bean to be within the range of other pulse crops [22]. In this study, the range in nickel content observed for Baki™ bean (1.49 to 8.71 ppm) is larger than the range reported by Ray et al. (2014) for pulses grown in Saskatchewan, CA (5.1 to 6.7 ppm). Furthermore, the mean value observed for Baki™ bean is lower than common bean (6.0 ppm) and higher than field pea (2.7 ppm) and lentil (1.5 ppm). They also quantified selenium in their study. The range observed for Baki™ bean is generally lower than the values they report for dry bean, chickpea, field pea, and lentil. Values at the higher end of the Baki™ bean range are most comparable to the selenium mean value reported for chickpea (0.732 ppm). We found that Baki™ bean had slightly higher selenium content than was reported for dry bean (0.443 ppm) and field pea (0.470 ppm), and less than the value reported for lentil (1.179 ppm) [33]. 

Careful attention to the tradeoff between yield and nutrient density is required by plant breeders who wish to develop crops with the potential to positively impact human health. For example, deficiencies occur in populations without access to nutrient dense crops, such as the epidemic of iron deficiency worldwide and the isolated and less common incidences of nickel and selenium deficiencies [34]. This has occurred in part due to the prioritization of yield in cereal crop breeding, resulting in increased carbohydrate content and a dilution of the micronutrients available [35].

#### 2.1.2. Folate

Folate, like iron, zinc, nickel, and selenium, is an essential micronutrient that must be obtained from the diet. Folate is a generic term for water-soluble B-complex vitamins, which exist in numerous chemical forms and function in single-carbon transfer reactions [36]. To obtain folate, populations must rely on staple foods or foods with added folate in the form of folic acid. Since 1981, the addition of 1.4 mg of folate per 1 kg grain in cereal grain products (e.g., pasta, bread, cereals) has been a mandated practice in the US [37,38]. Folate deficiency poses a significant risk to human health. Notably, a 13% reduction in odds of spina bifida, a type of neural tube defect, has been estimated for each 100 µg increase in daily dietary folate equivalent (DFE) consumed [39]. 

We found a wide range in DFE values for Baki™ Bean. Values ranged from approximately 2060 to 14,635 µg 100 g^−1^ (Table 1). Baki™ bean appears to have substantially higher folate content than other pulses. In an analysis of different pulse crop species and varieties grown at different locations in Saskatechewan, CA, the values reported by Jha et al. (2015) are generally much lower [40]. However, the lower end of the range observed for Baki™ bean overlapped with the higher end of the range of values reported for chickpea (3510 to 5890 µg 100 g^−1^) and common bean (1650 to 2320 µg 100 g^−1^). Comparatively, Baki™ bean may have higher content than lentil (1360 to 1820 µg 100 g^−1^) and pea (230 to 300 µg 100 g^−1^). They found significant variety and variety by location effects on folate content. Further research is needed to better understand how these factors may contribute to the variation observed for Baki™ bean in this study.

In general, pulses crops can play an important role in meeting recommended daily intakes of folate for adults (400 µg) and pregnant women (600 µg) when consumed regularly in modest amounts [38]. In the case of Baki™ bean, these results indicate that as little as 4 g of Baki™ bean would need to be consumed to provide the Recommended Daily Allowance (RDA). However, this assumes that all the folate present in raw Baki™ bean would be available for uptake. Certain compounds may negatively impact the bioavailability of folate [38]. The relationships between folate uptake and any inhibitory compounds in Baki™ bean remains unknown. Further research is needed to determine the bioavailability of folate in Baki™ bean and implications for the intake level of Baki™ bean to meet the RDA. 

#### 2.1.3. Canavanine

Canavanine is a non-protein amino acid associated with toxicity in plant and animal systems [41]. Canavanine can replace arginine in protein synthesis and cause autoimmunological diseases in humans or animals (e.g., systemic lupus erythematosus in humans). The rate of replacement may be increased in the presence of inflammation or other conditions that can lead to arginine deficiency [42]. Notably, consumption of canavanine containing seeds from *Hedysarum alpinum* is believed to have contributed to the death of Chris McCandless, the subject of the book *Into the Wild* [43]. Members of the genus *Canavalia* (e.g., *Canavalia ensiformis*; jack bean) accumulate canavanine in high concentrations, as does alfalfa or lucerne (*Medicago sativa*) to a lesser extent. Accumulation occurs primarily in seeds, where canavanine functions as a nitrogen storage resource [44]. Moreover, canavanine functions as an allelochemical and chemical barrier that can deter herbivory by insects and animals [45]. Germinated alfalfa seeds (i.e., sprouts) are one commercial source of canavanine in diets. While autoclaving can destroy canavanine in alfalfa seeds, it also renders them unviable [46]. 

Baki™ bean is derived from sainfoins (*Onobrychis* spp.), members of the Canavanine-accumulating clade within the Leguminosae family. Most of the agriculturally important legumes (e.g., alfalfa, pea, chickpea, soybean, clover, cow pea, mung bean, lentil, and fava bean) also occur in this clade [47]. Therefore, we analyzed Baki™ bean to determine if canavanine is present, which had not been previously reported to our knowledge. Each Baki™ bean sample contained canavanine below the limit of detection (Table 2). We found canavanine to occur in the alfalfa check. The concentration observed is lower than reported by Rosenthal et al. (2000) [48]. Moreover, canavanine was detected in the Baki™ bean positive control. The canavanine used to spike the sample was 98.60% recovered. These results suggest that Baki™ bean does not contain canavanine at detectable levels. Therefore, there is minimal to no risk of canavanine toxicity in Baki™ bean when present in the diets of animals and humans. 

### 2.2. Study II

Results are provided in Table 3, Table 4 and Table 5. 

#### 2.2.1. Macronutrients

Compared to other pulses, Baki™ bean has substantially higher protein content. In this study, protein content ranged from 29.61% to 39.55% (as-is basis). This is a slightly larger range than previously reported for Baki™ bean by Craine et al. (2023) [22]. Whole sainfoin seeds, with intact seed coats, generally contain 30–37% protein [49,50,51,52,53]. Baki™ bean also has slightly higher fat content than most pulses, and it is most comparable to chickpea. We found slightly lower fat content than Craine et al. (2023) reported [22]. Carbohydrate content is lower than in other pulses, and it is primarily comprised of dietary fiber as shown by Craine et al. (2023) [22]. The Baki™ bean samples analyzed in this study had consistently low moisture content, ranging from 7.10% to 7.85%.

#### 2.2.2. Mycotoxins

Mycotoxins are secondary metabolites produced by microfungi. They can cause illness and death in animals and humans when consumed in foods where they are present [54]. Postharvest moisture management is a critical step in managing mycotoxin levels [55]. The environment for current Baki™ bean seed production is arid, with annual precipitation occurring at less than 300 mm. Relative humidity and seed moisture at harvest are more easily managed in such environments. This likely contributed in part to the low levels of mycotoxins observed for Baki™ bean in this study (Table 4). Ochratoxin A content in one sample was above the detection limit. However, the value observed is below the limit for unprocessed cereals (5 ppb) established in European Union Commission Regulation (EC) No. 2023/915. A separate sample had fumonisin content above the detection limit, with a value below the recommended limit for corn products intended for human consumption (2 to 4 ppm) [56]. Vomitoxin content in Baki™ bean samples was below the 1 ppm restricted level established by the Food and Drug Administration for finished wheat products [57]. Because sainfoins are drought-tolerant and adapted to dry, arid environments with relatively low soil moisture availability, an expansion of Baki™ bean to novel production regions is likely to resemble the environment where the study samples were produced. Therefore, the prevalence and associated risk from mycotoxins will likely be minimal, posing little to no concern for the safety of Baki™ bean consumption for animals or humans. However, continuous monitoring of mycotoxins will ensure that Baki™ bean levels meet levels as established by relevant regulatory agencies.

#### 2.2.3. Microorganisms

Baki™ bean had low or undetectable levels of microorganisms. This is likely also a result of the production environment and seed producer’s postharvest processing practices. Coliforms, *Escherichia coli*, and mold occurred at levels below the detection limit, with the exception of one sample with a mold result of 30 colony forming units (cfu) g^−1^. Salmonella was not present in any samples. Standard plate counts ranged from 180 to 3300 cfu g^−1^, and yeast counts ranged from 20 to 30 cfu g^−1^. Specifications for microorganisms are generally established by companies or industry trade companies to ensure that proper production and handling practices are being observed to maintain health and safety. For example, the commercial item description authorized by the United States Department of Agriculture for precooked, dehydrated, refried, whole, and blended beans states the following: salmonella shall be negative; yeast and mold shall be less than 100 cfu g^−1^; and total plate counts shall be less than 100,000 cfu g^−1^ [58]. Using this example as a framework for specifications, the raw Baki™ bean samples included in this study are well below the limits specified. These results suggest that animals or humans consuming Baki™ bean are at minimal risk from microorganisms.

#### 2.2.4. Pesticides

There are a limited number of herbicides registered for use on sainfoins [18]. Nevertheless, the Baki™ bean samples were analyzed for a total of 230 pesticides. All analytes were found at levels below the respective limits of detection (Appendix A). Pesticide use and exposure will need to be monitored in Baki™ bean according to relevant regulatory mandates to ensure safety. 

## 3. Materials and Methods

### 3.1. Seed Production

For both investigations, Baki™ bean was sourced from commercial sainfoin seed producers in Montana, US. For the first study, seed was produced between 2018 and 2020 by three producers (i.e., A, B, C). Producer A provided a sample of the sainfoin varieties Eski [59] and Shoshone [60]. Producer B provided a sample of Shoshone, Delaney, and Rocky Mountain Remont. Producer C provided a sample of AAC Mountainview [61], Delaney, Remont, and Renumex [62]. Rocky Mountain Remont is a variety developed by selecting adapted plants from the variety Remont (<https://www.montanaseeds.com/about-us>, accessed on 15 September 2023). For additional information on certain sainfoin varieties, see USDA NRCS Plant Materials Technical Note No. MT-91 (<https://www.nrcs.usda.gov/plantmaterials/mtpmctn12043.pdf>, accessed on 15 September 2023). 

For the second study, producer C provided samples of seed harvested in 2022 from six seed production fields, representing the varieties AAC Mountainview, Delaney, Renumex, and Rocky Mountain Remont. For Delaney, samples were provided from three different production fields, differentiated by use as seed registered with the state of Montana’s Department of Agriculture’s Seed Program (Helena, MT, USA) (one field), and farmer-maintained seed (two separate fields).

### 3.2. Sample Preparation for Chemical Analyses

Seed was prepared for analysis as described previously [22,23]. Briefly, seeds were removed from pods and depodded seeds, with seed coats intact, and they were haphazardly selected from the fraction retained on a 2.778 mm sieve.

### 3.3. Chemical Analyses

For the first study, elemental analysis (USP <233>) and determination of folate (GPAL 03-HPLC-FA) was performed by Great Plains Analytical Laboratories (GPAL; Kansas City, MO, USA). Analysis of L-canavanine was performed by the University of Missouri-Columbia Agricultural Experiment Station Chemical Laboratories (Columbia, MO, USA). Briefly, 10 g of seed was ground through a 0.5 mm sieve and then rolled to mix. Approximately 100 mg of sample was weighed, and 1 mL of extraction solution, consisting of 80% methanol and 0.1 N HCl, was added. The mixture was vortexed and then sonicated for 20 min. Samples were extracted for 4 h and vortexed every hour to resuspend the sample. Each sample had 100 mcl of internal standard (IS) in 35% sulfosalicylic acid added to each sample. Protein was precipitated by centrifugation for 40 min at approximately 12,000 revolutions per minute at 4 °C. The supernatant was removed and mixed at a ratio of 1:1 with physiological buffer (e.g., PF 1 buffer) and filtered through a 0.2 micron filter. The filtered solution was transferred to a plastic vial and placed on a Hitachi (Tokyo, Japan) 8900 amino acid analyzer. A lithium buffer system and high-speed physiological amino acid column was used for amino acid separation of individual free amino acids and determination of canavanine content. All seed samples were analyzed alongside a positive control (i.e., Baki™ bean sample spike with L-canavanine standard), an alfalfa seed check, and a control sample (i.e., blank sample). The positive control was prepared by adding 14.5 mg of canavanine to 1008.1 mg of Baki™ bean (1.44 g 100 g^−1^) and by rolling to mix. A commercial seed sample was purchased and used for the alfalfa check (Country Creek Acres LLC (Chesterfield, MO, USA) Sprouting Seed, amazon.com). 

For the second study, all analyses were performed by GPAL. Proximate content was determined according to the American Association of Cereal Chemists International (AACCI) [63]. Analyses included moisture (method 44-15.02), ash (method 08-01.01), protein (determination of nitrogen × 6.25; method 46-30.01), and fat (Acid Hydrolysis; method 30-10.01). Content of fungal and microbial contaminants were determined according to the Association of Official Agricultural Chemists (AOAC) [64]. Analyses included standard plate count (method 990.12), coliforms (petrifilm; method 991.14), *Escherichia coli* (petrifilm; method 991.14), and salmonella (method 2009.03). Analyses performed via enzyme-linked immunosorbent assay (ELISA) (Veratox^®^ Kits, Neogen^®^, Lansing, MI, USA) include included Aflatoxin (method RI 050901), Fumonisin (method OMA 2001.06), Ochratoxin A, T2 Toxin, Vomitoxin (method RI 090901), Zearalenone. Yeast and mold were determined according to the Food and Drug Administration (FDA) Bacteriological Analytical Manual (BAM) Chapter 18 [65]. A mutliclass pesticide screen (231 analytes) was conducted according to United States Department of Agriculture Chemistry Laboratory Guidebook Pesticide Screening Test (USDA CLG PST5) [66]. Carbohydrates were calculated by difference. A summary of analytes by study with methods and references is provided in Appendix A.

### 3.4. Statistical Analyses

Elemental data and folate were adjusted to a dry matter basis. All other data were reported on an as-is basis, unless otherwise noted. When results from heavy metal analytical testing were returned below the respective limit of detection (LOD), the result was adjusted to half of the LOD value to allow for further statistical analysis.

All statistical analyses were conducted using the R statistical software (version 4.3.1) unless otherwise noted [67]. Summary statistics (e.g., count, mean, standard deviation) were generated using the summarize function [68] or base R functions. Heavy metal mean values were compared to maximum levels for pulses, or other relevant foods if a limit was not provided for pulses, in the Codex Alimentarius International Food Standards [69] using a one-sided, one-sample *t*-test with a null hypothesis stating that there is no difference between the sample distribution mean and the corresponding maximum limit. 

## 4. Conclusions

We provide a preliminary investigation into the safety of Baki™ bean from two separate and complementary studies. In the first study, Baki™ bean had undetectable levels of arsenic, beryllium, chromium, lead, mercury, and silver. Barium, cadmium, nickel, and selenium were present at levels comparable to the range reported for other crops and food products and likely do not pose a safety concern. Baki™ bean did not contain canavanine, compared to an alfalfa seed check. The second study corroborates past studies, demonstrating a relatively high protein and fat content, low carbohydrate content, and similar ash content of Baki™ bean compared to other pulse crops. Aflatoxin, T2 Toxin, and Zearalenone were not detected. Ochratoxin A, Fumonisins, and Vomitoxin were present at levels below regulatory thresholds. Salmonella, coliforms, and *E. coli* were not detected. Results of standard plate counts, yeast, and mold were below the detection limits in certain samples and occurred at low levels when present in other samples. A total of 230 pesticides were screened for and were not found to be present at levels above the respective detection limits. Overall, the combined analyses did not reveal a substantial cause for concern to health and safety. These studies provide preliminary evidence supporting the potential use of Baki™ bean as a novel pulse crop for human consumption. This work will support further studies, which are necessary to provide a more robust determination of safety and quality. 

## Figures and Tables

**Table 1 molecules-29-01777-t001:** Content of heavy metals and folate for the Baki™ bean samples (*n* = 9), reported on a dry matter basis.

Analyte	LOD	Mean	Standard Deviation	Minimum	Maximum
Arsenic (ppm)	0.10	<0.10	NA	<0.10	<0.10
Barium (ppm)	0.50	3.49	3.44	1.13	9.6
Beryllium (ppm)	0.050	<0.050	0	<0.050	<0.050
Cadmium (ppm)	0.020	0.07	0.05	0.03	0.15
Chromium (ppm)	1.00	<1.00	NA	<1.00	<1.00
Lead (ppm)	0.10	<0.10	NA	<0.10	<0.10
Mercury (ppm)	0.010	<0.010	NA	<0.010	<0.010
Nickel (ppm)	1.00	3.79	2.87	1.49	8.71
Selenium (ppm)	0.50	0.07	0.21	0	0.64
Silver (ppm)	0.50	<0.50	NA	<0.50	<0.50
Folate (DFE) (µg)	44	10,038.75	5175.86	2059.72	14,634.96

LOD: limit of detection; NA: not applicable; ppm: parts per million; DFE: dietary folate equivalents.

**Table 2 molecules-29-01777-t002:** Canavanine content of Baki™ bean, in addition to alfalfa seed, control (i.e., blank), and positive control (i.e., Baki™ bean sample spiked with canavanine standard). Values are reported as is, on a 100 g^−1^ sample basis.

Analyte	Statistic	Alfalfa	Control	Positive Control	Baki™ Bean
	*n*	1	1	1	15
Canavanine (g 100 g^−1^)	Mean	0.8	<0.01	1.42	<0.01
	Standard Deviation	NA	NA	NA	NA

Detection limit: <0.01; NA: not applicable.

**Table 3 molecules-29-01777-t003:** Quantification of macronutrients in Baki™ bean samples.

Parameter	Units	LOD	*n*	Mean	Minimum	Maximum	Standard Deviation
Ash	%	0.020	3	3.42	3.29	3.58	0.10
Carbohydrates	%	NA	6	48.44	45.91	55.69	3.65
Fat	%	0.10	6	5.15	3.54	8.36	1.70
Moisture	%	0.20	6	7.52	7.10	7.85	0.31
Protein	%	0.20	6	35.47	29.61	39.55	3.49

LOD: limit of detection; NA: not applicable.

**Table 4 molecules-29-01777-t004:** Quantification of mycotoxins in Baki™ bean samples.

Parameter	Units	LOD	*n*	Mean	Min	Max	Standard Deviation
Aflatoxin	ppb	5.00	6	<5.00	<5.00	<5.00	NA
Fumonisin	ppm	0.50	3	<0.50	<0.50	0.5	NA
Ochratoxin A	ppb	2.00	6	<2.00	<2.00	2.9	NA
T2 Toxin	ppb	50.00	3	<50.00	<50.00	<50.00	NA
Vomitoxin	ppm	0.025	3	0.210	0.162	0.298	0.076
Zearalenone	ppb	50.00	3	<50.00	<50.00	<50.00	NA

LOD: limit of detection; ppm: parts per million; ppb: parts per billion; NA: not available.

**Table 5 molecules-29-01777-t005:** Quantification of microorganisms in Baki™ bean samples.

Parameter	Units	LOD	*n*	Mean	Min	Max	Standard Deviation
Coliforms	cfu/g	10	3	<10	<10	<10	NA
*Escherichia coli*	cfu/g	10	3	<10	<10	<10	NA
Mold	cfu/g	10	3	<10	<10	30	NA
Salmonella	cfu/25 g	NA	3	Negative	Negative	Negative	NA
Standard Plate Count	cfu/g	10	3	1370	180	3300	1687
Yeast	cfu/g	10	3	25	20	30	7

LOD: limit of detection; cfu: colony forming units; NA: not available.

## Data Availability

Data can be made available upon request at sainfoinconsortium.org.

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
