# Peer review of "Perennial Baki™ Bean Safety for Human Consumption: Evidence from an Analysis of Heavy Metals, Folate, Canavanine, Mycotoxins, Microorganisms and Pesticides"

_molecules, 2024, doi:10.3390/molecules29081777_

Round 1

Reviewer 1 Report

Comments and Suggestions for Authors

Review of manuscript: "Perennial Baki™ Bean Safety for Human Consumption: Evidence from Analysis of Heavy Metals, Folate, Canavanine, Mycotoxins, Microorganisms, and Pesticides"

The reviewed manuscript is a continuation of an earlier publication and is considerably poorer than its predecessor.
Authors investigated representative lots of seed from commercial producers and determined the content of folate, specific elements (e.g., heavy metals), and L-canavanine, a non-proteinogenic amino acid found in certain leguminous plants. How was the material selected to be representative? How was the randomness of the sample maintained? This is extremely important in statistical analysis.
Authors investigated seed produced from a single producer in a single year, representing different varieties and production locations, and determined the content of mycotoxins, microorganisms, and pesticides, in addition to macronutrients. This clearly indicates a two-way experience. Appropriate statistical methods are required to analyze such an experiment. However, they were not used in the reviewed manuscript! Why? It should be supplemented with the results of two-way analysis of variance. Information on the effects of individual differential factors should be provided. In this type of experiment, it is very important to evaluate genotype-environment interactions. It, too, was missing. Without the above analyses, the manuscript is not suitable for publication in any scientific journal.

What about the interdependence of the observed characteristics? They are undoubtedly dependent. Authors completely omitted these issues.
"3.4 Statistical Analyses": This section should list all the statistical methods used in achieving the research objective. Listing the software is not a description of the methodology!!!

Table 1: SD=0 indicates that all observations are identical. With the number of observations n=9 is unlikely and impossible. It should be checked.

Table 1: "SD", Table 2: "STD" - Explain the differences in defining these two parameters.

Table 2: STD=0 for N=15???!!!

"n" or "N"?

Table 4: STD=NA. Why? After all, observations of three or six have been made. Calculate and provide the values.
Table 5: STD=NA. Why? After all, observations of ten have been made. Calculate and provide the values.

The method of citation does not comply with the journal's policy.

Paper needs major revision.

Author Response

Please find point-by-point responses in bold below.

The reviewed manuscript is a continuation of an earlier publication and is considerably poorer than its predecessor.

Authors investigated representative lots of seed from commercial producers and determined the content of folate, specific elements (e.g., heavy metals), and L-canavanine, a non-proteinogenic amino acid found in certain leguminous plants. How was the material selected to be representative? How was the randomness of the sample maintained? This is extremely important in statistical analysis.

We were limited in our ability to control the randomness and representativeness of the sample, as the collection was done in collaboration with the farmer. We are trusting that each farmer used appropriate sample techniques to provide a sample representative of their seed lot. Since the farmer collected and provided the sample, we could only control how a subsample was collected from the larger sample provided by the farmer. In collecting the subsample, seed from the farmer provided sample was haphazardly selecting seeds as mentioned at the end of section 3.2. The referenced studies in 3.2 provide additional information on the samples preparation. If any of the additional information provided here, or in the referenced studies, would be worthwhile to include, please advise.

Authors investigated seed produced from a single producer in a single year, representing different varieties and production locations, and determined the content of mycotoxins, microorganisms, and pesticides, in addition to macronutrients. This clearly indicates a two-way experience. Appropriate statistical methods are required to analyze such an experiment. However, they were not used in the reviewed manuscript! Why? It should be supplemented with the results of two-way analysis of variance. Information on the effects of individual differential factors should be provided. In this type of experiment, it is very important to evaluate genotype-environment interactions. It, too, was missing. Without the above analyses, the manuscript is not suitable for publication in any scientific journal.

We agree that genetic and environmental factors that could influence seed components should be investigated. However, this was not the focus of these studies and we believe that the experimental design does not support such analyses. These studies were intended to determine if the compounds analyzed were present, and at what levels, in seed samples available from different farmers. Production is limited, as were the resources available for seed testing, resulting in limited replication for the various factor combinations (year, variety, location). These studies serve as a necessary preliminary analysis, from which future studies could be conducted to build on the results presented. Even without the analyses suggested, which are beyond the scope of this manuscript, we believe that this manuscript represents an important contribution of novel information currently lacking in the literature. Furthermore, We believe that this study will benefit future studies that can be designed to provide an in-depth analysis of genotype-environment interactions and other factors that could influence the results.

To clarify and articulate these points to the reader, we have added additional sentences to the end of section 1 and to the conclusion and removed mention of “all statistical analyses” to clarify that there are no additional analyses beyond the summary statistics section 3.4. 

What about the interdependence of the observed characteristics? They are undoubtedly dependent. Authors completely omitted these issues.

Could this comment be clarified to help our consideration.

"3.4 Statistical Analyses": This section should list all the statistical methods used in achieving the research objective. Listing the software is not a description of the methodology!!!

Section 3.4 includes all of the statistical analysis performed. Summary statistics (i.e., descriptive statistics) were used to report the central tendency and dispersion of the results from the chemical analyses, and were calculated using the R statistical software and function listed. 

Table 1: SD=0 indicates that all observations are identical. With the number of observations n=9 is unlikely and impossible. It should be checked.

This is an error as reported. We have changed this value to NA not applicable, because all samples had values below the limit of detection.

Table 1: "SD", Table 2: "STD" - Explain the differences in defining these two parameters.

This is an error. STD is now used in Table 1. 

Table 2: STD=0 for N=15???!!!

This is an error as reported. We have changed this value to NA not applicable, because all samples had values below the limit of detection.

"n" or "N"?

We have replaced N and N with n to present the sample population.

Table 4: STD=NA. Why? After all, observations of three or six have been made. Calculate and provide the values.

All samples had values below the respective limits of detection. We don’t believe that it is possible to calculate and provide a standard deviation in this situation, and maintain the use of NA or not applicable. 

Table 5: STD=NA. Why? After all, observations of ten have been made. Calculate and provide the values.

All samples had values below the respective limits of detection. We don’t believe that it is possible to calculate and provide a standard deviation in this situation, and maintain the use of not applicable (NA). 

The method of citation does not comply with the journal's policy.

In-text citations have been updated with square brackets according to the Molecules citation style. 

Paper needs major revision.

We appreciate the reviewers' time in reviewing the manuscript and providing comments.

Reviewer 2 Report

Comments and Suggestions for Authors

The manuscript presents an evaluation of the chemical and microbiological safety of a perennial crop alternative to the most common cereals produced year by year. The work is interesting in view of how agriculture is being affected by climate change, which also affects the concentrations of contaminants in the agricultural products. However, the work needs some revision.

Introduction: the authors should better explain the main characteristics of a perennial crop and highlight the difference with annual cropping systems. This could better contextualize the work

Line 30: What does "volatile weather events" mean?

Elemental analysis: it is well-known that the concentrations of cadmium and inorganic arsenic in rice are strongly dependent on the cultivation procedures. It is not clear if different types of cultivation can affect the contamination of Baki beans as well as it is not clear if the soil composition can influence the concentration of heavy metals. A mention to these aspects should be made

Line 188: replace "thank" with "than"

Line 203: please, note that Commissione Regulation (EC) 1881/2006 has been replaced by Commission Regulation (EU) 2023/915

Section 3 should be moved before the section on results and discussion (section 2)

Paragraph 3.4 It is not clear for what extent the statistical analysis has been applied (t-test? p-value? or...?)

Author Response

Please find point-by-point response in bold below.

The manuscript presents an evaluation of the chemical and microbiological safety of a perennial crop alternative to the most common cereals produced year by year. The work is interesting in view of how agriculture is being affected by climate change, which also affects the concentrations of contaminants in the agricultural products. However, the work needs some revision.

Introduction: the authors should better explain the main characteristics of a perennial crop and highlight the difference with annual cropping systems. This could better contextualize the work

Line 30: What does "volatile weather events" mean?

We used volatile here in relation to weather events that are likely to change suddenly and unexpectedly. 

Elemental analysis: it is well-known that the concentrations of cadmium and inorganic arsenic in rice are strongly dependent on the cultivation procedures. It is not clear if different types of cultivation can affect the contamination of Baki beans as well as it is not clear if the soil composition can influence the concentration of heavy metals. A mention to these aspects should be made

We appreciate this suggestion and have included this as a caveat to consider, both in the context of rice and Baki beans. 

Line 188: replace "thank" with "than"

This mistake has been corrected. 

Line 203: please, note that Commissione Regulation (EC) 1881/2006 has been replaced by Commission Regulation (EU) 2023/915

We appreciate the attention to this detail and have updated the manuscript accordingly. 

Section 3 should be moved before the section on results and discussion (section 2)

Paragraph 3.4 It is not clear for what extent the statistical analysis has been applied (t-test? p-value? or...?)

Summary (i.e., descriptive) statistics were used to report the central tendency and dispersion of the results from the chemical analyses, as mentioned in section 3.4. The focus of these studies was to determine if the compounds analyzed were present, and at what levels, in seed samples available from different farmers. Therefore, the statistical analyses are limited to this objective. We added additional sentences at the end of section 1 to articulate this to the reader, and have removed mention of “all statistical analyses” in section 3.4. to clarify that there are no additional analyses beyond the summary statistics. 

Round 2

Reviewer 1 Report

Comments and Suggestions for Authors

The answers provided by the Authors of the manuscript show that:

- They are unsure about the randomization of the sample, which most likely indicates a lack of it.

- Not using a two-way ANOVA for a two-factor experiment is incorrect. Whatever the purpose of the work, the analyses should be done correctly! Why are such answers not provided by the authors in purely agronomic aspects? Because it would be incorrect.

- The Authors have no idea about the principles of relationships between traits, and they undertake to describe them. It is imperative that they contact a statistician.

- Section 3.4. is still not a methodological description, but an indication of the software. It should be supplemented with, among other things, testing the consistency of the empirical distribution with the normal distribution.

- The abbreviation STD was not explained. There is no such abbreviation in statistics.
- There is no point in taking just one measurement (Table 2). It is necessary to supplement knowledge with basic assumptions from experimentation.

- The "N" was not replaced with an "n" everywhere. Sometimes there are double markings, which looks very strange.

- By far the majority of measurements are below the detection threshold. This manuscript makes no sense at all. It is unlikely to follow a normal distribution, which rules out the possibility of any testing.
- Citations still do not comply with the journal's policy.

Paper needs major revision.

Author Response

- They are unsure about the randomization of the sample, which most likely indicates a lack of it.

The two studies were designed to investigate whether the target compounds are present in Baki™ bean and at what levels. This survey was intended to sample the population of seed produced commercially in the US. Samples from seed lots were requested from seed companies and provided by companies as available. The authors could not control the various factors of seed production due to this opportunistic sampling approach. Therefore, the samples represent various cultivars, locations, and years. If this additional information does not address this comment relating to the randomization of the sample, please clarify so we can provide additional information as necessary. 

- Not using a two-way ANOVA for a two-factor experiment is incorrect. Whatever the purpose of the work, the analyses should be done correctly! Why are such answers not provided by the authors in purely agronomic aspects? Because it would be incorrect.

The authors wish to reiterate that this manuscript includes two studies designed to investigate whether the target compounds are present in Baki™ bean and at what levels. As mentioned in the previous comment and section 3.1 Seed Production, seed samples were sourced from commercial seed growers. The authors did not manipulate the production of the seed samples through an experimental design. The lack of an experimental design prohibits the ability of the authors to adequately account for the various factors that could influence seed production. For example, the samples were sourced from growers in different geographies, using different production practices to produce seed from different varieties of different stand ages in different years. Therefore, the authors assert that there is not adequate factorial replication to support the suggested analysis. In consultation with a statistician, as suggested by the reviewer, the statistician concurs with the authors’ previous statement on the feasibility of a two-way ANOVA given structure of the data. From consulting with a statistician, we have included a series of t-tests to test if the sample distribution mean differs significantly from the published maximum level for each heavy metal, where available. We agree that agronomic factors are important to consider. Future studies will further investigate the compounds that were found to be present in this preliminary study to quantify which factors contribute to the variation observed.  

- The Authors have no idea about the principles of relationships between traits, and they undertake to describe them. It is imperative that they contact a statistician.

In contacting a statistician, clarification is needed to better understand what is meant by the relationships between traits. These two studies were designed to investigate whether the target compounds are present in Baki™ bean and at what levels. This request is possibly beyond the scope of the study as designed, due to the sampling design used. 

- Section 3.4. is still not a methodological description, but an indication of the software. It should be supplemented with, among other things, testing the consistency of the empirical distribution with the normal distribution.

The authors have included methodology for the t-tests now included. 

- The abbreviation STD was not explained. There is no such abbreviation in statistics.

It appears that track changes in the document may have caused confusion. To avoid further confusion, standard deviation is now represented explicitly in each table. 

- There is no point in taking just one measurement (Table 2). It is necessary to supplement knowledge with basic assumptions from experimentation.

The authors relied on the collaborating analytical chemistry laboratory to implement a standardized and appropriate testing methodology. Therefore, the laboratory included three controls to compare the samples to. These controls included a negative, and two positives (alfalfa check and spiked).

- The "N" was not replaced with an "n" everywhere. Sometimes there are double markings, which looks very strange.

It appears that track changes in the document may have caused confusion.The authors kindly suggest reviewing the document version without track changes to avoid possible confusion. 

- By far the majority of measurements are below the detection threshold. This manuscript makes no sense at all. It is unlikely to follow a normal distribution, which rules out the possibility of any testing.

To reiterate, the two studies were designed to investigate whether the target compounds are present in Baki™ bean and at what levels. As a result, some compounds were not detected and are considered absent at an appreciable level. When found to be absent, the authors believe that few, if any, additional analyses are possible or required. However, for certain compounds (e.g., cadmium), certain samples had values below the limit of detection (absent) while others had measurable amounts (present). In this situation, the values below the limit of detection were adjusted to a real value to allow for additional statistical analysis. This methodology is now included in section 3.4 Statistical Analyses. Please see lines 287 - 297 for additional information. 

- Citations still do not comply with the journal's policy.

Author and year have been added where necessary in-text, in addition to the previously included citation in [XX] format (see line 174). If this change does not adequately address the reviewer comment, clarification regarding how the citations are noncompliant with the journal’s policy would be appreciated. 

Paper needs major revision.

We appreciate the feedback and opportunity to improve the manuscript. We have made major revisions as suggested.

Reviewer 2 Report

Comments and Suggestions for Authors

All my comments have been addressed

Author Response

All my comments have been addressed

We appreciate the feedback and opportunity to improve the manuscript.